# Inducible Knockout of 14-3-3β Attenuates Proliferation and Spheroid Formation in a Human Glioblastoma Cell Line U87MG

**DOI:** 10.3390/brainsci13060868

**Published:** 2023-05-27

**Authors:** Kellie Gallo, Bhairavi Srinageshwar, Avery Ward, Carlos Diola, Gary Dunbar, Julien Rossignol, Jesse Bakke

**Affiliations:** 1Biochemistry, Cellular, and Molecular Biology, College of Science and Engineering, Central Michigan University, Mount Pleasant, MI 48559, USA; 2Program of Neuroscience, Central Michigan University, Mount Pleasant, MI 48859, USA; 3Foundational Sciences Department, College of Medicine, Central Michigan University, Mount Pleasant, MI 48859, USA; 4Department of Psychology, Central Michigan University, Mount Pleasant, MI 48859, USA

**Keywords:** glioblastoma, YWHAB, 14-3-3β, tumor spheroid, Tet-on CRISPR

## Abstract

Glioblastomas (GBs) are the most common and malignant brain tumors in adults. A protein encoded by the gene YWHAB, 14-3-3β, is commonly found to be upregulated throughout the initiation and progression of GB. The 14-3-3β has oncogenic roles in several different types of cancer cells through interactions with proteins such as Bad, FBI1, Raf-1, Cdc25b, and others. Previous RNA interference studies have shown that 14-3-3β promotes proliferation, cell cycle progression, and migration and invasion of GB cells. However, despite the many oncogenic functions of 14-3-3β, a CRISPR/Cas9 knockout model of 14-3-3β has not been investigated. This study confirmed previous findings and showed that siRNA inhibition of 14-3-3β results in reduced cellular proliferation in a human glioblastoma cell line, U87MG. We also used a YWHAB Tet-On CRISPR/Cas9 U87MG cell line that, upon doxycycline induction, leads to robust Cas9 expression and subsequent knockout of 14-3-3β. Using this model, we show that loss of 14-3-3β significantly reduces cellular proliferation and spheroid formation of U87MG cells.

## 1. Introduction

Glioblastomas (GBs) are one of the most common and malignant brain tumors in the central nervous system [1]. Although the incidence is only 3.19 per 100,000 persons in the United States [2], the mortality rate is high [3], with a five-year survival rate of approximately 5.0% [1]. Treatment for GB typically includes a combination of neurosurgery, chemotherapy, and radiotherapy. Despite some treatment advancements, the survival rate remains extremely low due to recurrence following surgical resection, chemotherapy resistance [4], and lack of targeted therapies [5].

The 14-3-3 proteins are a family of highly conserved acidic polypeptides found primarily in the mammalian brain consisting of seven isoforms (β, γ, η, τ, σ, ε, and ζ). The 14-3-3 family influences a wide variety of cellular functions, including cell signaling, regulation of cell cycle progression, and apoptosis by functioning as molecular adapters, blocking binding sites, or modifying protein conformation of their targets [6,7]. All seven of these isoforms are upregulated in a variety of cancers, but 14-3-3β is one of the most common of the subunits upregulated in GB [8].

The 14-3-3β (encoded by the gene YWHAB) is commonly upregulated and associated with increased malignancy and poorer prognosis in glioblastoma [8]. In recent studies, antisense 14-3-3β sequences decreased the cell motility of U87MG cells, a glioblastoma cell line commonly used in research [9]. Additionally, the knockdown of 14-3-3β in U87MG cells using RNAi reduced cellular proliferation and migration [10] and induced senescence [11]. In this study, we investigated the phenotype of a complete knockout of 14-3-3β in U87MG cells using an inducible Tet-on CRISPR-Cas system. Constitutive reduction of 14-3-3β protein, as most studies to date have done, often results in a population of cells with minimal reduction of 14-3-3β due to its role in growth and survival. In other words, cells with minimal reduction would out-compete the cells with the most reduction of 14-3-3β. Thus, this system has several advantages over previous studies, which primarily include avoiding proliferation defects caused by constitutive and prolonged 14-3-3β protein reduction; rather, we can grow the cells normally and then induce knockout with doxycycline for terminal experiments.

## 2. Materials and Methods

### 2.1. Cell Culture

The human glioblastoma-derived cell line, U87MG-Luc2 (ATCC HTB-14-LUC2), was maintained at 37 °C, 95% humidity, and 5% CO_2_ in Minimum Essential Media with Earle’s salts and L-glutamine (Gibco, Waltham, MA, USA) supplemented with 10% FetalPlex serum GeminiBio, Sacramento, CA, USA). The human embryonic 293T cells (ATCC CRL-3216) and Lenti-X 293T cells (Takara, Shiga, Japan) were maintained at 37 °C, 95% humidity, and 5% CO_2_ in Dulbecco’s Modified Eagle’s medium with 4.5 g/L D-glucose and L-glutamine (DMEM, Gibco., Waltham, MA, USA), 10% fetalPlex serum (GeminiBio, Sacramento, CA, USA), and 1 mM sodium pyruvate. U87MG-Luc2, 293T, and Lenti-X 293T were authenticated with Short Tandem Repeat (STR) markers, and cells are routinely PCR tested for mycoplasma (ABM, Richmond, BC, Canada). Cells were passaged using 0.25% Trypsin-EDTA (Sigma Aldrich, St. Louis, MO, USA).

### 2.2. siRNA Transfection into U87MG Cells

Cells were seeded at 2.4 × 10^5^ cells per well in 6-well plates 24 h prior to transfection. Two hours before transfection, media was removed and replaced with fresh-complete media. The 25 nM of YWHAB siRNA or 25 nM Cyclophilin B siRNA (Horizon Discovery, Waterbeach, UK) was transfected into U87MG-Luc2 cells (ATCC HTB-14-LUC2) using RNAiMAX (Thermo Fisher Scientific, Waltham, MA, USA). After a 12 h transfection, media was removed and replaced with fresh-complete media.

### 2.3. siRNA and Inducible CRISPR Cell Proliferation Assays

Following transfection of YWHAB and Cyclophilin B control siRNA using RNAiMAX, in U87MG cells, cells were counted, and 1 × 10^5^ cells were seeded into 60 mm plates. Cells in triplicate were detached and counted on days 3 and 5 using both a hemocytometer as well as an automatic cell counter (Bio-Rad TC20; BioRad, Hercules, CA, USA).

Wild-type (WT) U87MG and U87MG cells stably expressing Tet-Cas9 YWHAB gRNA1 were seeded into eight 6-well plates at a density of 20,000 cells/well. A total of 1 mg/mL doxycycline final concentration (Invitrogen, Carlsbad, CA, USA) was added to twelve wells of each cell type. Three replicates were collected for each cell type and media condition (±doxycycline) on days 2, 4, 6, and 8. At each time point, cells were detached, collected, and counted using both a hemocytometer as well as an automatic cell counter (Bio-Rad TC20; BioRad, Hercules, CA, USA).

### 2.4. pCAG-eCas9-GFP-U6-gRNAYWHAB sgRNA Plasmid Construct

Three YWHAB sgRNAs were designed using online programs, E-CRISP [12] and CHOPCHOP [13,14,15], and subcloned into pCAG-eCas9-GFP-U6-gRNA vector (a gift from Jizhong Zou; Addgene plasmid # 79145) with an EGFP reporter as well as overhangs compatible for cloning into the Bbs1 site following digestion with Bbs1 restriction enzyme (Thermo Scientific, Waltham, MA, USA, ER1011). Complementary oligos of three YWHAB target sequences (Table 1) were heated at 37 °C for 30 min, 95 °C for 5 min, and annealed by decreasing the temperature 1 °C/min to 25 °C using a thermocycler (BioRad C1000; BioRad, Hercules, CA, USA). Next, the short double-stranded DNA fragments were ligated into the linearized Bbs1 site of the pCAG-eCas9-GFP-U6-gRNA vector. Ligation products were transformed into Escherichia coli (*E. coli*) NEB 5-alpha competent cells (New England Biolabs, Ipswich, MA, USA) and grown in LB (Luria–Bertani) medium containing 100 µg/mL ampicillin (Sigma–Aldrich, St. Louis, MO, USA) as a growth-selective marker. The cloned constructs were then purified using a ZR Plasmid Miniprep-Classic kit, sequenced (Sanger Sequencing), and analyzed using SnapGene software (from GSL Biotech; V6.0 snapgene.com, accessed on March 15, 2021; Boston, MA, USA) to confirm correct sgRNA insertion into the Bbs1 site.

### 2.5. Tet-On CRISPR-Cas Plasmids; Assembly, Lentivirus Generation, and Viral Transduction

The Lenti-X Tet-On 3G CRISPR/Cas9 system was purchased from Takara (Takara, Shiga, Japan) and used to create the doxycycline-inducible knockouts of YWHAB (14-3-3β protein). These cells are serially transduced, and after each viral transduction, a single-cell clone is verified and expanded. Parent plasmid maps are available and detailed on the manufacturer’s website (Takara, Shiga, Japan). Briefly, pLVX-EF1a-tet-3G plasmid encodes for the selectable marker G418, and the tet promoter, pLVX-TRE3G-Cas9 encodes for the selectable marker puromycin and Cas9, and pLVX-hyg-sgRNA encodes for the gRNA of interest (Figure 3A). sgRNA 1 (5′ ACACCCAATTCGTCTTGGTC 3′) and a non-targeting sgRNA were cloned into the pLVX-hyg-sgRNA plasmid. Following this, the Lenti-X Packaging Single Shots (Takara, Shiga, Japan) were used to produce individual lentiviruses for pLVX-EF1a-tet-3G, pLVX-TRE3G-Cas9, and pLVX-hyg-sgRNA vectors (Takara, Shiga, Japan) in Lenti-X 293T cells (Takara, Shiga, Japan) following the manufacturer’s protocol. Lentiviral titers were measured 48 and 72 h later using the Lenti-X GoStix (Takara, Shiga, Japan) and frozen in 1 mL aliquots at −80 °C. Next, we transduced 2 × 10^5^ U87MG-Luc2 cells in 6-well plates with the pLVX-EF1a-Tet3G vector using 4 µg/mL polybrene (Sigma–Aldrich, St. Louis, MO, USA). After 12 h, we replaced it with fresh media and incubated the cells at 37 °C for 48 h. Following the 48-h incubation, the cells were split into 3 × 10 cm dishes with fresh growth media. A total of 48 h after splitting (96 h post-transduction), we selected 800 µg/mL G418 (Sigma–Aldrich, St. Louis, MO, USA) for three weeks. We repeated this process, transducing in pLVX-TRE3G-Cas9-puro followed by dual selection with 400 µg/mL G418 and 2 µg/mL puromycin (Sigma–Aldrich, St. Louis, MO, USA) for one week. Following selection, colonies were chosen using cloning cylinders (Sigma–Aldrich, St. Louis, MO, USA) and transferred and grown out individually in 24-well plates. Several individual clones were assessed for doxycycline-induced Cas9 expression by comparing wells without doxycycline (Invitrogen, Carlsbad, CA, USA) to wells with 1 mg/mL doxycycline in triplicate. Quantitative PCR (methods described below) of Cas9 mRNA was used to select the cell clone with the biggest fold change of mRNA expression (referred to as gRNA 1), and this clone was then transduced with pLVX-hyg-sgRNA1 containing sgRNA 1 and pLVX-hyg-sgRNA1 containing the non-targeting gRNA sequence. These clonal Tet-Cas9-sgRNA cells were selected with 200 µg/mL hygromycin (Sigma–Aldrich, St. Louis, MO, USA). These cells were expanded and frozen in the vapor phase of liquid nitrogen for future experiments.

### 2.6. Lipofectamine Transfection of U87MG Cells

One day prior to transfection, cells were seeded in a 6-well plate at a cell density of 3.5 × 10^5^ cells per well. The following day, the media was replaced with fresh DMEM, and cells were transfected with pCAG-eCas9-GFP-U6-gRNA vector using Lipofectamine 3000 (Thermo Fisher Scientific, Waltham, MA, USA) according to the manufacturer’s protocol.

### 2.7. Quantitative PCR

RNA was extracted using the RNeasy kit (Qiagen, Hilden, Germany), and cDNA was synthesized using Superscript CellsDirect cDNA synthesis kit (Invitrogen, Carlsbad, CA, USA) according to the manufacturer’s protocol. Next, 50 ng of template cDNA, TaqMan Fast Advanced Master Mix (Thermo Fisher Scientific, Waltham, MA, USA), and TaqMan probes for YWHAB (Thermo Fisher Scientific, Waltham, MA, USA) and 18 s rRNA (Thermo Fisher Scientific, Waltham, MA, USA) were used to assess YWHAB mRNA expression.

Additionally, qPCR was done using SYBR green to detect fold change differences in Cas9 expression in the presence and absence of 1 mg/mL doxycycline (Invitrogen, Carlsbad, CA, USA). Following RNA extraction using the RNeasy kit (Qiagen, Hilden, Germany), the qPCR master mix was made using 12.5 µL of 2X TB Green Premix Ex Taq II (Tli RNase Plus), ROX Plus (Takara, Shiga, Japan), 2.5 µL of 4 mM Cas9 qPCR Primer Mix (Takara, Shiga, Japan), 0.125 µL of PrimeScript RT (Takara, Shiga, Japan), diluted to 20 U/mL in 2X TB Green Premix Ex Taq II (Tli RNase Plus), ROX Plus, 0.625 µL of Recombinant RNase Inhibitor (Takara, Shiga, Japan), and 2 µL of RNA sample (0.01 mg/mL), brought up to 25 µL total volume using nuclease-free water. The Cas9 qPCR Primer Mix was: Forward 5′ ACTACAAGGTGCCGAGCAAAA 3′ and Reverse 5′ CGCCAATGAGGTTCTTCTTTATGCT 3′.

Both TaqMan and SYBR green-based detection methods utilized the StepOnePlus (Thermo Fisher Scientific, Waltham, MA, USA) PCR system.

### 2.8. Western Blot Analysis

Doxycycline was added to a final concentration of 1 mg/mL to U87MG-Luc2 cells containing the Tet-Cas9-sgRNA inserts and incubated at 37 °C for five days. Cells were then lysed with Western-ReadyTM Rapid Protein Extraction buffer (BioLegend, San Diego, CA, USA). Next, cells were centrifuged at 16,000× *g* for 10 min at 4 °C, and the supernatant was transferred to a new tube. The 2X Tris-Glycine SDS Sample Buffer (Thermo Fisher Scientific, Waltham, MA, USA) and NuPAGE Sample Reducing Agent (Invitrogen, Waltham, MA) were added to the samples prior to loading on a 10% SDS-PAGE. Following electrophoresis, proteins were transferred at 300 milliAmps onto a nitrocellulose membrane (BioRad, Hercules, CA, USA) for 2 h. The membrane was blocked at room temperature for 1 h in SuperBlock Dry Blend (TBS) Blocking Buffer (Thermo Fisher Scientific, Waltham, MA, USA). The membrane was incubated at 4 °C overnight with rabbit monoclonal anti-14-3-3 antibody (1:1000; RRID AB_2809697, Thermo Fisher Scientific, Waltham, MA, USA) or mouse monoclonal anti-beta-tubulin antibody (1:1000; RRID AB_2565030, BioLegend, San Diego, CA, USA). After washing, the membrane was incubated with either Donkey anti-Rabbit antibody (1:10,000; cat no. RRID AB_10953628, LI-COR Biosciences, Lincoln, NE, USA) or Donkey anti-Mouse antibody (1:10,000; cat no. RRID AB_10956166, LI-COR Biosciences, Lincoln, NE, USA) at room temperature for 90 min. Protein expression was quantified using LI-COR ImageStudio Version 5.2 software (LI-COR Biosciences, Lincoln, NE, USA).

### 2.9. 3D-Spheroid Formation Assay

A 3D-spheroid formation assay was done as previously described [16] with some changes; rather than inferring spheroid volume from diameter, we directly measured the total cells within the spheroid. Briefly, Wild type (WT) U87MG and U87MG cells stably expressing Tet-Cas9 YWHAB gRNA1 (known in Figure 4 as YWHAB gRNA U87MG) were seeded into a round-bottom ultra-low adhesion 96-well plate at a density of 1500 cells/well. There were 12 replicates of WT U87MG cells, 12 replicates of WT U87MG with 1 mg/mL of doxycycline (Invitrogen, Carlsbad, CA, USA), 12 replicates of YWHAB gRNA U87MG cells, and 12 replicates of YWHAB gRNA U87MG cells with 1 mg/mL of doxycycline. The cell culture medium was changed every 2–3 days, and on day 7 the spheroid media was aspirated, and spheroids were subject to 3–4 min of 40 μL TrypLE (Thermo Fisher Scientific, Waltham, MA, USA). Then, 200 μL of fresh media was added, and the entirety of the mixture was transferred to a centrifuge tube. The cells were centrifuged at 300 g for 5 min, and the media was removed. The cell pellets were resuspended in 300 μL of a PBS/Trypan Blue (BioRad, Hercules, CA, USA) mixture (1:1), and live cells were counted using both a hemocytometer well as an automatic cell counter (BioRad TC20; BioRad, Hercules, CA, USA).

### 2.10. Statistics

All statistics were done using GraphPad Prism Version 9 software (San Diego, CA, USA). Two-way ANOVA multiple comparisons with Tukey post-hoc test (alpha 0.05) or Holm-Sidak multiple t-test (alpha 0.05) were used to determine data significance.

## 3. Results

### 3.1. siRNA Knockdown of 14-3-3β Reduces Proliferation of U87MG Cells

We knocked down 14-3-3β protein with siRNA and observed a significant reduction of YWHAB mRNA (Figure 1A) as well as 14-3-3β protein (Figure 1B). Consistent with previous studies [10,11], we were able to show a significant reduction in the proliferation of U87MG cells following siRNA knockdown of 14-3-3β compared to wild-type (no treatment) and control siRNA (cyclophilin B) U87MG cells.

### 3.2. CRISPR gRNA Selection

We utilized two online gRNA design tools (E-CRISP and CHOPCHOP) to pick several candidate gRNAs that target YWHAB. We sub-cloned three YWHAB gRNAs into a constitutively expressed CAS9 plasmid, pCAG-eCas9-GFP-U6-gRNA, and we transiently transfected these three YWHAB targeting plasmids into U87MG cells. Utilizing PCR primers targeting the gRNA cut sites, the three YWHAB gRNAs were screened for knockout efficiency, and two of the three YWHAB gRNAs showed a significant reduction of YWHAB mRNA with gRNA 1, demonstrating the largest reduction (Figure 2A). The difference among these clones is typical and likely a result of the gRNA sequence performance but could also be due to transfection variation. We validated gRNA1 by western blot analysis of the YWHAB protein product, 14-3-3β, and found that it was significantly reduced in this U87MG pooled cell population (Figure 2B).

### 3.3. Tet-On CRISPR YWHAB gRNA Validation

Since long-term cultures with cytotoxic or anti-proliferative genes are difficult, we decided to use an inducible system to study the effects of 14-3-3β knockout. To that end, we utilized a three-plasmid system (Figure 3A and for additional details, see methods). We engineered a Tet-On CRISPR/CAS9 U87MG cells that induce Cas9 expression only upon the addition of doxycycline, and when doxycycline is removed, Cas9 is no longer expressed. This allows for temporal regulation of Cas9 and limits editing to a short period, allowing the cells to proliferate as normal prior to editing, unlike transfection of YWHAB siRNA, which has immediate proliferative defects. These cells were clonally selected after each viral transduction (Figure 3A) to achieve the greatest fold change of Cas9 following the addition of 1 mg/mL doxycycline (Figure 3B). Clone 1 had the greatest induction of Cas9. The following transduction to express YWHAB gRNA1 14-3-3ß was undetectable. Several cell populations (P1-4) were selected, clonally expanded, and then split for comparison following the addition of doxycycline. U87MG YWHAB sgRNA P1-P4, as well as the parental cell line, Tet-Cas9 U87MG, were analyzed for 14-3-3β protein following a five-day media treatment without doxycycline or with 1 mg/mL of doxycycline (Figure 3C). We found that induction of Cas9 by doxycycline resulted in a significant reduction of 14-3-3β protein among the YWHAB sgRNA populations (Figure 3D,E).

**Figure 3 brainsci-13-00868-f003:**
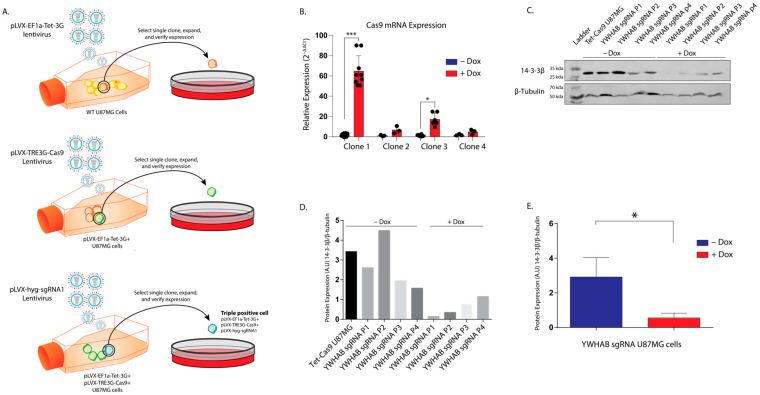
**A.** Generation of a Tet-inducible CRISPR knockout of 14-3-3β in U87MG cells. (**A**) Relative expression (2^−∆∆Ct^) of Cas9 expression in stably expressing Tet-CAS9 clonal U87MG cells upon doxycycline (1 mg/mL) administration for 48 h. (**B**) Quantification of the western blot analysis. *** denotes *p* < 0.001 and * denotes *p* < 0.01 (Holm-Sidak multiple *t*-test). (**C**) Western blot analysis of 14-3-3β and β-tubulin expression in parental clone1 Tet-Cas9 U87MG cells, as well as Tet-Cas9 YWHAB sgRNA U87MG cells of differing populations (P1, P2, P3, P4). (**D**) Comparison of the Tet-Cas9 YWHAB sgRNA populations with and without the addition of doxycycline (1 mg/mL for 72 h). (**E**) Average 14-3-3β protein expression of the YWHAB sgRNA U87MG cell populations (P1, P2, P3, and P4) with ±1 mg/ml doxycycline. * denotes *p* < 0.05 (Holm Sidak multiple *t*-tests).

### 3.4. Inducible CRISPR Knockout of 14-3-3β Reduces Proliferation and 3D Spheroid Formation of U87MG Cells

We analyzed the proliferation rate of Tet-On CRISPR/CAS YWHAB gRNA U87MG cells and found that induction of Cas9 by doxycycline and the resulting reduction of 14-3-3β protein results in a significant reduction in U87MG cell proliferation (Figure 4A), in agreement with our siRNA results. To mimic in vivo tumor growth, we used a popular 3D-spheroid formation assay to grow cells as a spheroid on round bottom ultra-low adhesion plates. We found that the knockout of 14-3-3β in U87MG cells results in a significant reduction in spheroid cell number (Figure 4B). This result is specific to the knockout of 14-3-3β, as doxycycline alone or YWHAB gRNA expressing U87MG cells without doxycycline do not have the same growth defect.

**Figure 4 brainsci-13-00868-f004:**
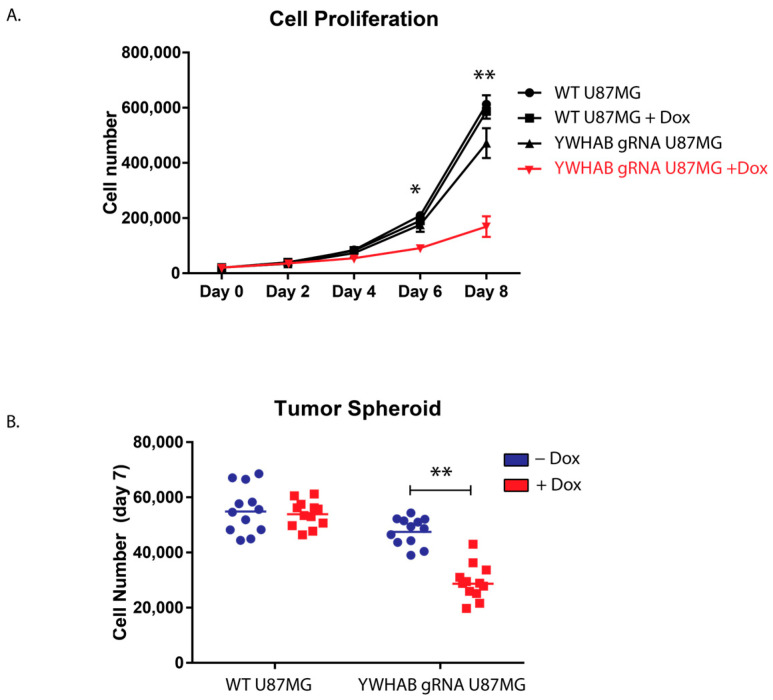
Tet-inducible CRISPR knockout of 14-3-3β reduces U87MG cell proliferation and tumor spheroid growth in vitro. (**A**) Wild-type (WT) U87MG cells and Tet-Cas9 YWHAB gRNA U87MG cells were plated in replicates of 12 with and without doxycycline (1 mg/mL), and three experimental replicates were counted at days 2, 4, 6, and 8. * denotes *p* < 0.05 ** denotes *p* < 0.01 (Tukey two-way ANOVA). (**B**) Wild-type (WT) U87MG cells and Tet-Cas9 YWHAB gRNA U87MG cells were plated on 96-well ultra-low adhesion round bottom plates in replicates of 12 with and without doxycycline (1 mg/mL). The first group (on the left) shows WT U87MG cells with ±1 mg/mL doxycycline, and the second group (on the right) shows YWHAB gRNA expressing U87MG cells with ±1 mg/ml doxycycline YWHAB gRNA U87MG cells with doxycycline would be the group with knockout for 14-3-3β. The cells were counted on day 7; cells were plated on day 0, and doxycycline (if applicable) was also added on day 0. ** denotes *p* < 0.01 (Tukey two-way ANOVA).

## 4. Discussion

The 14-3-3 proteins are a family of highly conserved phospho-binding proteins that are most commonly found in the brain. The 14-3-3 proteins regulate many essential cellular functions8. There are seven different isoforms (β, γ, ε, η, ζ, σ, and τ/θ) encoded by genes YWHA(B-Z) [17]. Each of the 14-3-3 subunits regulates its targets by blocking protein binding sites, modifying protein conformation, or acting as molecular adapters [7]. As a result, this protein family influences a large number of protein kinases, phosphatases, and signaling pathways, playing important roles in mitogenic signal transduction, apoptosis, and cell cycle control [17]. All seven isoforms are found to be upregulated in various types of cancer, but the upregulation of 14-3-3β and η subunits is most commonly associated with GB [8].

The 14-3-3β has been shown to mediate apoptosis, cell proliferation, and angiogenesis. Upregulation of 14-3-3β results in increased binding and phosphorylation of Bad, which in turn disrupts the binding of the antiapoptotic factor Bcl-2 and suppression of apoptosis [18]. Additionally, Raf-1/MAPK activity has been found to be positively correlated to the cellular expression of 14-3-3β [19]. Additionally, the expression of VEGF is correlated with 14-3-3β expression. Forced expression of antisense 14-3-3β siRNA in AFB1-induced rat hepatoma K1 and K2 cells resulted in the downregulation of VEGF mRNA expression and decreased angiogenesis, while the frequency of apoptosis increased [20].

More recent studies investigating the effects of 14-3-3β knockdown in A172 and U87MG cells have shown an increase in tumor suppressor p27, a known CDK inhibitor that arrests the cell cycle at G1 by inactivating E- and D- cyclins, which occurred due to the decreased phosphorylation of Erk. Depletion of 14-3-3β not only increased the expression of tumor suppressor p27 but also decreased phosphorylation of AKT, leading to increased senescence in GB cells [11]. The increased senescence upon silencing of 14-3-3β allows for activation of cell cycle arrest in response to the DNA damage present in GB cells. An additional knockdown of 14-3-3β using RNAi in U373-MG cells showed a decrease in proliferation and invasion/migration as well as an increase in apoptosis [10]. Our siRNA results confirmed these previously published findings. We also found that silencing of YWHAB using siRNA results in a decrease of U87MG-Luc2 growth in vitro when mRNA and protein expression levels of 14-3-3β were decreased (Figure 1). However, none of these studies included the complete removal of 14-3-3β. The residual 14-3-3β protein after RNA interference may still confer some survival benefit. Indeed, CRISPR screens have identified many more essential genes compared to RNAi screens, most likely due to the complete knockout of the gene [21,22].

RNA interference studies only partially remove the protein of interest, so in order to assess the effect of the complete removal of 14-3-3β, we generated U87MG cells that constitutively express Cas9 and gRNAs targeting 14-3-3β (Figure 2). Consistent with previous studies [10,11], these cells fail to proliferate, and a large percentage of the population dies (data not shown). This was an expected result based on 14-3-3β’s role in cell survival and our own data showing a reduction in proliferation following 14-3-3β RNA interference. Additionally, as data reproducibility is vital to the scientific method, we opted to include those results within this publication as well.

To avoid the proliferative defects of constitutive 14-3-3β knockout, we developed a tetracycline-inducible CRISPR/Cas U87MG cell line. This would allow us to grow the cells normally and control the timing of the editing of YWHAB. We generated a clonal Tet-Cas9 U87MG cell line that expressed very low levels of Cas9 mRNA in the absence of doxycycline and sufficient levels of Cas9 following the addition of doxycycline to a final concentration of 1 mg/mL. Doxycycline binds to the Tet promoter with higher affinity than tetracycline and is the inducing drug of choice for Tet-on or off studies. The Tet-Cas9 clone we chose had a 74-fold change in Cas9 mRNA levels between cells without doxycycline and cells with doxycycline (Figure 3A).

We also assessed the growth of 14-3-3β inducible CRISPR/Cas U87MG cell line using two experimental approaches. First, to assess proliferation, we used two-dimensional growth on coated flasks and observed a similar reduction of proliferation as in our siRNA studies. Upon doxycycline induction of YWHAB knockout, the cells rapidly began to slow proliferation (Figure 4A). Additionally, to assess the role of 14-3-3ß in tumor formation, we conducted 3D tumor spheroid studies utilizing round bottom ultra-low adhesion plates [16,23]. After induction of YWHAB knockout, the U87MG spheroid contains significantly fewer cells, again likely through defects in cellular proliferation (Figure 4B). We hypothesize that the effect would be much greater, but despite utilizing an engineered clonal cell line that has robust Cas9 expression following doxycycline induction, we still have cells within the population that have residual protein expression due to incomplete knockout. This is not unexpected and is supported by western blot analysis in Figure 3C. While some animal studies have been completed utilizing xenotransplantation methods10, genetic knockout studies are needed to investigate 14-3-3β’s role in GB tumor initiation, proliferation, and treatment resistance.

## 5. Conclusions

We have shown, through the use of a novel doxycycline-inducible CRISPR system, that knockout of 14-3-3β results in both impaired proliferation and decreased cells within a 3D-spheroid of U87MG cells, a human glioblastoma cell line. We also confirmed previously published RNA interference studies showing a defect in GB cellular proliferation, and reproducibility of data is integral to the scientific method [24]. Taken together, we provide evidence that further testing in patient-derived xenograft (PDX) cell lines and animal studies is warranted to investigate the therapeutic potential of knocking out 14-3-3β, as well as to investigate 14-3-3β’s role in the initiation and progression of glioblastoma. Completion of both studies is critical prior to translating these findings into the clinic. The Tet-ON inducible CRISPR knockout of 14-3-3β is likely more clinically relevant than constitutive knockdown as it may be more representative of temporal drug inhibition of 14-3-3β or even future targeted gene therapy.

## Figures and Tables

**Figure 1 brainsci-13-00868-f001:**
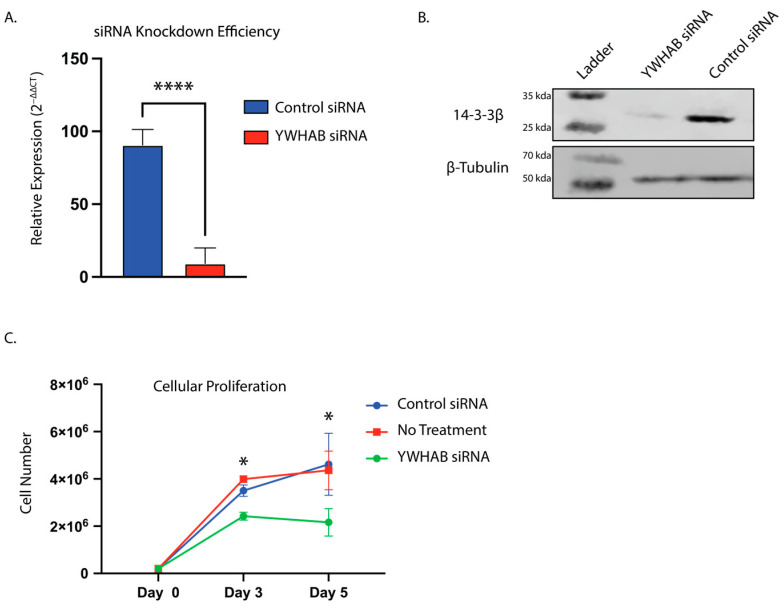
siRNA knockdown of 14-3-3β reduces U87MG cell proliferation. (**A**) Knockdown efficiency of 14-3-3β (YWHAB) compared to control (cyclophilin B) siRNA transfected with RNAiMAX and using YWHAB and 18 s TaqMan probes to detect mRNA expression. **** denotes *p*-value of <0.001 (student’s *t*-test). (**B**) Western blot analysis showed a significant reduction in YWHAB (28 kDa) protein expression compared to control siRNA (cyclophilin b) transfected using RNAiMAX. β-tubulin was used as a loading control. (**C**) Cell proliferation of U87MG cells following transfection of YWHAB (14-3-3β) or control (cyclophilin B) siRNA compared to non-treated cells U87MG cells. Cells, in triplicate, were counted on days three and five for each treatment. * denotes *p*-value of <0.05 (Tukey two-way ANOVA).

**Figure 2 brainsci-13-00868-f002:**
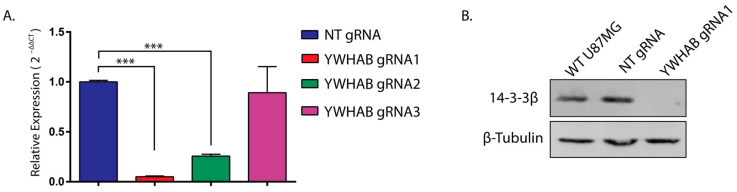
Transient CRISPR Knockout of 14-3-3β. (**A**) Relative expression (2^−∆∆Ct^) of U87MG cells transfected with three different YWHAB gRNAs. *** denotes *p* < 0.001 (Holm-Sidak multiple *t*-tests). (**B**) Western blot analysis showing protein expression of 14-3-3β and β-tubulin in WT U87MG cells, U87MG cells transiently transfected with non-targeting gRNA (NT), and U87MG cells transiently transfected with YWHAB gRNA1.

**Table 1 brainsci-13-00868-t001:** Complements and Reverse Compliments for Three sgRNA Sequences for Deletion of YWHAB.

	Protospacer Sequence	Reverse Complement of Protospacer Sequence
sgRNA-1	5′ ACACCCAATTCGTCTTGGTC 3′	5′ GACCAAGACGAATTGGGTGTC 3′
sgRNA-2	5′ CTGCTGGGAGTTCGACACAG 3′	5′ CTGTGTCGAACTCCCAGCAGC 3′
sgRNA-3	5′ CACTGTGTCGAACTCCCAG 3′	5′ GCTGGGAGTTCGACACAGTGC 3′

## Data Availability

All data is contained within the article.

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
