# Peer review of "Inducible Knockout of 14-3-3β Attenuates Proliferation and Spheroid Formation in a Human Glioblastoma Cell Line U87MG"

_brainsci, 2023, doi:10.3390/brainsci13060868_

Round 1
Reviewer 1 Report
Very interesting manuscript and study that addresses a significant clinical issue. The methods are well-described, state-of-the-art, and sound. I have a few minor suggestions that would improve the paper.
Line 62: reference 8 at end of sentence was not formatted correctly.
Lines 69-72: The last sentence at the end of the Introduction should be rewritten. It is unclear what the phrase beginning "...which includes the avoidance of proliferation defects outside of terminal experiments..." means.
Section 2.9: It would be great to provide an image or cartoon rendering of a 3D spheroid if possible. This could either be in the methods or in the results with Figure 4b.
Line 218: references 10,11 not formatted correctly.
Lines 372-373: Was size of spheroid measured? It is not included in results. If spheroid size is a function of cell number, this statement should refer to this dependent variable. However, if spheroid size is independent of cell number, this should be reported and interpreted.
As a reader, I wanted to hear why the three clones tested provided the different results in Figure 2A. Would the authors like to speculate about this?
I would also have liked to hear the authors' address any potential issues related to translating their findings to the clinic, even if it is a brief statement.
Author Response
We would like to thank the reviewer for the submitted comments/questions and hopefully addressed them in the attached file and in the revised manuscript.

Reviewer 2 Report
This topic is very interesting, but some points need to be revised:
- Lines 69-72: "This system has several advantages over previous studies, which includes the avoidance of proliferation defects outside of terminal experiments and the complete and prolonged removal of the 14-3-3β protein" - But what is the purpose of this paper? please improve this point.
- Lines 294-296: "We also conducted in vitro 3D spheroid tumor studies and found that the knockout of 14-3-3β in U87MG cells results in a significant reduction in spheroid cell number (Fig. 4B)." What do authors mean? Improve this results report. Discuss more figure 4 and improve figure legend.
- Lines 335-337: "More recent studies investigating the effects of 14-3-3β knockdown in A172 and U-87 MG cells have shown an increase in tumor suppressor p27, a known CDK inhibitor that... " Consider at these very recent and interesting papers: -- doi: 10.3390/neurolint15020037 -- doi: 10.3390/genes14020501 -- doi: 10.3389/fimmu.2018.00727
- Lines 384-385: ""We also confirmed previously published RNA interference studies, and reproducibility of data is integral to the scientific method [24]". this point must be discussed in the discussion section, as previous results.
- Conclusion should be improved. Report some results there.
Minor editing of English language required
Author Response

(The authors gave the same response as above.)

Reviewer 3 Report
Gallo et al. here found an interesting and potentially important method for suppressing cell proliferation and tumor spheroid in human glioblastoma cell lines. Through RNA interference, constitutive knock-out and inducible knock-out, they confirmed persistent or temporal loss of key molecule 14-3-3beta efficiently suppressed tumorigenesis. The data provided in the manuscript is solid and supported the arguments they made.
N/A
Author Response

(The authors gave the same response as above.)

Round 2
Reviewer 2 Report
Good
Author Response
Thank you